Habitat selection of black grouse in an isolated population in northern Germany—the importance of mixing dry and wet habitats

Tost Daniel 1
Ludwig Tobias 1
Strauss Egbert 1
Jung Klaus 2
Siebert Ursula ursula.siebert@tiho-hannover.de 1
1 Institute for Terrestrial and Aquatic Wildlife Research, University of Veterinary Medicine Hannover, Foundation , Hannover , Germany
2 Institute for Animal Breeding and Genetics, University of Veterinary Medicine Hannover, Foundation , Hannover , Germany
Roper James
Electronic publication date: 2022 Oct 17
Publication date: 2022
Volume: 10
Electronic Location ID: e14161
Received 2022 Mar 9; Accepted 2022 Sep 9
Copyright: ©2022 Tost et al.
Copyright year: 2022
Copyright holder: Tost et al.
License: This is an open access article distributed under the terms of the Creative Commons Attribution License, which permits unrestricted use, distribution, reproduction and adaptation in any medium and for any purpose provided that it is properly attributed. For attribution, the original author(s), title, publication source (PeerJ) and either DOI or URL of the article must be cited.
License URL: https://creativecommons.org/licenses/by/4.0/

Keywords: Grouse, Habitat selection, Conservation science, Habitat suitability model, Tetrao tetrix

Funding: Ministry of Food, Agriculture and Consumer Protection of Lower Saxony DFG, German Research Foundation) 491094227 University of Veterinary Medicine Hannover, Foundation The project was funded by the Ministry of Food, Agriculture and Consumer Protection of Lower Saxony. This Open Access publication was funded by the Deutsche Forschungsgemeinschaft (DFG, German Research Foundation) - 491094227 ”Open Access Publication Funding” and the University of Veterinary Medicine Hannover, Foundation. There was no additional external funding received for this study. The funders had no role in study design, data collection and analysis, decision to publish, or preparation of the manuscript.

==============================
Wildlife habitats in general must provide foraging, hiding and resting places as well as sites for reproduction. Little is known about habitat selection of black grouse in the lowlands of Central Europe. We investigated habitat selection of seven radio tagged birds in an open heath and grassland area surrounded by dense pine forests in the northern German Lüneburg Heath Nature Reserve. This site carries one of the last remaining populations in the Central European lowlands. Using resource selection functions based on presence/background data, we estimated the probability of black grouse occurrence by availability of, or distance to habitat types as well as vegetation diversity indices. Black grouse preferred undisturbed and heterogeneous habitats far from dense forests with wide sand heaths, natural grasslands and intermixed bogs, diverse vegetation and food sources, low density of (loose) shrub formations and solitary trees. Wetlands were extremely important in a landscape that is dominated by dry heaths and grasslands. About 4% (9 km2) of the nature reserve was a suitable habitat for black grouse, mostly due to lack of open areas due to the amount of dense forest, and because smaller, open heaths are only partly suitable. We suggest that to improve habitat quality and quantity for the grouse, habitat patch size and connectivity must be increased, along with a mosaic of heterogeneous landscape structures in these habitat islands. Our results may be used to inform and improve black grouse habitat management in the region and elsewhere.

Introduction

Black grouse (Tetrao tetrix) numbers have been declining for decades in northern Central Europe, and many local populations are extinct due to large-scale habitat loss (Ludwig, Storch & Gärtner, 2009; Ludwig, Storch & Graf, 2009; Ludwig, Storch & Wübbenhorst, 2008; Segelbacher et al., 2014). Although the black grouse is listed as least concern by the IUCN Red List (IUCN, 2016), it is listed as endangered in Germany and critically endangered in the northern German federal state of Lower Saxony (Krüger & Sandkühler, 2022). There, in Lower Saxony, the last remaining population is in the Lüneburg Heath region (Strauß et al., 2018). This autochthonous black grouse population is fragmented into five subpopulations distributed among the Lüneburg Heath Nature Reserve and four neighbouring areas used for military purposes (Strauß et al., 2018). The core areas are surrounded by intensively used agricultural and large contiguous forestry areas (Cordes et al., 1997) and distances between the core areas are 7 to 15 km (Strauß et al., 2018). Open heath and natural grassland areas that are expected to be potential black grouse habitats in the five core areas cover each between 13 and 86 km2 (totalling 197 km2) (Strauß et al., 2018). Despite different protection and biotope improvement measures, such as continuous maintenance of the heath biotopes, removal of regenerating young trees, predator control and visitor guidance (Cordes et al., 1997; Kaiser, 2015), the population could not be permanently stabilized at a level of above 200 individuals in the past 20 years (Strauß et al., 2018). After a minimum population size in 1999 with 142 individuals, the population temporarily increased to 261 by 2011 and then declined again to a historic minimum of 126 individuals by 2020 (Strauß et al., 2018; Tost et al., 2020).

In its main distribution area of the Eurasian Palaearctic, black grouse inhabit transition zones between forests and raised bogs and in Asia also of steppes, while old, dense forests are usually excluded from their habitat (Glutz von Blotzheim, Bauer & Bezzel, 1994). Whether in the boreal zone, alpine, upland, or lowland regions, (semi-)open habitats with light-loving dwarf and berry shrubs are preferred (Klaus et al., 1990; Patthey et al., 2011). Being adapted to dynamic environments with changing mosaics of different succession stages, grouse are sensitive to anthropogenically homogenized, disturbed habitats (Angelstam, 2004; Ludwig, Storch & Wübbenhorst, 2008; Immitzer, Nopp-Mayr & Zohmann, 2014). Where found elsewhere in Europe, black grouse habitats are geographically and scenically very different from those in the Lüneburg Heath, with those in the Netherlands and England being the most comparable (Baines, 1994).

Black grouse seem to have sufficient food in all seasons and individuals are healthy in the Lüneburg Heath Nature Reserve (Strauß et al., 2018; Strauss et al., 2022), so it is unclear why the population does not grow. Furthermore, the population seems to have partly adapted to anthropogenic influences, such as recreational use, that cause fragmentation and loss of available habitats (Tost et al., 2020). Predation pressure and changing weather conditions may be important for population decline (Wegge & Kastdalen, 2008). However, habitat quality and quantity are the main prerequisites for stable populations (Klaus et al., 1990) but there are few studies that address habitat selection by black grouse (Baines, 1994; Immitzer, Nopp-Mayr & Zohmann, 2014; Patthey et al., 2011; Schweiger, Nopp-Mayr & Zohmann, 2012; White, Warren & Baines, 2015) and provide management implications that are applicable for the Lüneburg Heath habitats based on landscape characteristics.

We therefore investigated habitat selection (Boyce et al., 2002; Forester, Im & Rathouz, 2009; Northrup et al., 2013) and habitat suitability (Hirzel & Le Lay, 2008) of black grouse in the Lüneburg Heath. Generalized linear regression models were calculated using presence and random-background data (Phillips et al., 2009) and environmental data from large-scale vegetation and habitat mapping within the Lüneburg Heath Nature Reserve. Model results from the study area were used to spatially predict habitat availability and distribution for the entire nature reserve. These predictions of habitat suitability can then be used to develop or improve conservation strategies for black grouse in the nature reserve and possibly in the neighboring military training areas.

We wished to determine (1) how important different habitats are for black grouse, (2) which of the available habitats are used and which of these are preferred. Additionally, we wished to examine the importance of small-scale wetlands and structurally diverse habitats for black grouse in this landscape.

Materials & Methods

Study area

The study area (16.3 km2) encompassed the home ranges of seven individually tagged grouse within the 235 km2 Lüneburg Heath Nature Reserve in Lower Saxony, Germany (53.167930°N, 9.939770°E). The reserve is composed of 66% forest, 22% heath and grassland, 6% arable land, 5% pasture and 1% paths, buildings and water bodies (Cordes et al., 1997; Kaiser, 2015; Strauß et al., 2018). Its core areas consist mainly of open heath and natural grassland with small-scattered shrub formations, juniper or pioneer vegetation and is surrounded by dense pine forests. This landscape in the North German lowland is characterized by sandy ground and terminal moraines with flat undulating relief. The nature reserve’s heathlands are the remains of a historic agricultural landscape development (Tost et al., 2020), the preservation of which is realized today by modern mechanized landscape management, sheep farming, and heath burning (Kaiser, 2015). The reserve is located in the transition zone between Atlantic and continental climate, with an average annual precipitation of 806 mm and average annual temperature of 9.4 °C (CDC, 2021). This study focuses solely on the Lüneburg Heath Nature Reserve; the neighbouring black grouse habitats in areas used for military purposes are not considered.

Black grouse data

About one quarter of the northern German black grouse population is located in the open heath and grassland of the nature reserve (Strauß et al., 2018). In 2007, 78 individuals (45 cocks; 33 hens) were counted during annual censuses conducted by the foundation Stiftung Naturschutzpark Lüneburger Heide and the Lower Saxony Federal Ornithological Station. Since then, numbers have been falling (Tost et al., 2020), with 66 individuals (38; 28) in 2011 and 60 (27; 33) in 2012. In 2021 only 33 individuals (22; 11) were counted in the nature reserve (F Stucke, 2021, pers. comm.).

Data were collected as previously described in Tost et al. (2020). A total of seven birds were captured and fitted with backpack mounted, battery operated GPS tags (e-obs GmbH, Gruenwald, Germany) in the eastern part of the nature reserve in 2011 and 2012. Five cocks were fitted with 38 g, and two hens with 28 g tags. One cock was caught and tagged in May 2011, the remaining six birds (four cocks and two hens) were caught and tagged between March and May 2012 (Table 1). Both hens reproduced successfully, with the first clutch of one hen preyed and a second clutch hatched. Five birds were lost to predators, one bird (ID 1101) went missing and could not be recovered and one bird’s fate (ID 1205) remains unknown after the tag’s battery was depleted in December 2012 (Table 1). GPS-locations were taken every three hours between 01:00 and 22:00 daily during the entire sampling period. In total 2,296 locations were taken. The minimum number of locations per individual was 96, the maximum was 546. The observation time ranged between 61 and 222 days. All stages of the animal experiment were conducted under a permit from the Lower Saxony Institute for Consumer Protection and Food Safety (LAVES, Dept. 33 Animal Welfare, permit number: 33.9-42502-04-11/0364). All field experiments were approved by the lower nature conservation authorites of the district Soltau-Fallingbostel (permit number: 09.509 N 24 - Lü 2 - 4) and district Harburg (permit number: 71 21/1.2.1−0.0 - 2011-0081 -Kr).

Table 1 Summary of tagged black grouse individuals including number of GPS locations, date of capture and duration of data collection.

Home range size was calculated as 95% kernel.

Animal ID	Sex	Number of locations	Date tagged	Last position	Home range (ha)	Age	Weight (g)	
1101	m	159	08-05-2011	12-07-2011	39	adult	1,304	
1201	m	452	25-03-2012	08-09-2012	133	adult	1,361	
1202	m	408	01-04-2012	10-09-2012	51	adult	1,365	
1204	m	199	02-05-2012	02-07-2012	198	adult	1,287	
1205	f	436	04-05-2012	03-12-2012	98	adult	947	
1206	f	546	06-05-2012	15-12-2012	192	adult	991	
1207	m	96	09-05-2012	30-07-2012	78	yearling	1,189	

Environmental data

Habitat data were obtained from vegetation and biotope mapping carried out by Kaiser & Purps (2012) on behalf of the state of Lower Saxony for the baseline survey of the NATURA 2000 area. These data were kindly provided by the Lower Saxony Water Management, Coastal Defence and Nature Conservation Agency (NLWKN) for our analyses but may not be made publicly available in the original. Mapping was carried out between 2009 and 2011 according to the specifications of the Lower Saxony mapping key (von Drachenfels, 2004) in the open heath areas. As most important variables mapping included species inventories of flora and fauna, vertical vegetation layers (stratification) and horizontal proportion of plant cover, (bio-) geological parameters as well as types of land use. Based on this, biotopes (Table S1) including their subunits, additional features and their state of conservation were recorded. However, data are missing for parts of the forested areas of the nature reserve. Based on digital aerial photographs and field surveys the large-scale heaths were subdivided into 5,698 polygons with a total area of 59.7 km2 by biotope composition, and then the plots were inventoried during detailed mappings (Kaiser & Purps, 2012). Due to the size of the area, some of the surveys were conducted quantitatively along representative transects by recording percentages of the lengths of transect sections with homogeneous biotope characteristics. From this, plant species cover percentages per polygon were derived (Kaiser & Purps, 2012).

Data processing

Data processing was performed in R version 4.1.2 (R Core Team, 2021). Modeling of habitat selection was based on the habitat and vegetation mappings, which were available as vector data. The data contained 166 different biotope types and their subunits. These were partly grouped within their hierarchical order to reduce the number of independent model variables in the further process (Table S1). Subunits of biotopes were partially grouped at the level of their main units, or biotopes (main units) were grouped at the level of their super units (e.g., ‘degraded raised bogs still capable of natural regeneration’ (MP) and ‘transition mires and quaking bogs’ (MW), etc. were grouped to ’raised bogs, mires, and fens’ (bog)). Hierarchical grouping was done according to the Lower Saxony mapping key (von Drachenfels, 2004), which can be translated into the habitat types of the EU Habitats Directive (92/43/EEC), and resulted in ten groups of biotopes, eight of which were used for modelling (Table 2). In addition, three diversity indices were calculated using the mapping inventory of 511 plant species at polygon level (summed quantity of food plant cover (Strauß et al., 2018), number of plant species, Shannon index of plant species diversity, coefficient of variation of vegetation). The edited vector data were then rasterized into 10 × 10-meter cells for each grouped biotope separately and coded as present (1)/absent (0). To account for heterogeneous edge effects, we fanned out habitat layers using focal weight with a radius of 25 m, converting the binary data to continuous values between 0 and 1 (Hijmans, 2021). In case of sparsely distributed or apparently unused structures distance rasters were created instead (Table 2).

In the next step, background points (n = 14,000) were spatially randomly distributed (Hijmans et al., 2021) within the home ranges of the seven tagged birds plus a buffer radius of 603 m, which was calculated as the 95th percentile of step distances of successive GPS positions (Boyce, 2006). This served to provide a clear spatial boundary for the model area based on realistic movement distances of black grouse rather than an arbitrarily defined reference area (Senay, Worner & Ikeda, 2013; VanDerWal et al., 2009). The data table for model building was then generated by extracting the raster values of all biotope and diversity index layers at the background points and telemetry locations.

Table 2 Habitat and diversity variables.

Type	Variable	Variable description	
Distance to habitat type (km)	dinf	Distance to infrastructure (settlement, roads, trails)	
dw	Distance to forest	
dp	Distance to pastures (grassland)	
dd	Distance to open soil (dunes)	
Availability of habitat type	hc	Sand heaths	
ng	Natural grasslands	
bog	Raised bogs, mires, and fens	
shr	shrub formations (solitary trees and scattered shrub)	
Diversity indices	fpc	Summed quantity of food plant cover	
nveg	Number of plant species	
H’	Shannon Index of plant diversity	
Notes.

Availability of habitat is given as proportion of coverage of grouped habitats (between 0 and 1), and diversity indices are dimensionless.

Data analysis of habitat selection

Habitat selection was analysed using generalized linear mixed-effects models (R package lme4) with logistic regression and binomial error structure (Bates et al., 2015; Kuznetsova, Brockhoff & Christensen, 2017), where the dependent variable was composed of GPS-locations (presence) and random background points (Phillips et al., 2009) as binary response (Brotons et al., 2004). To account for repeated measurements, individuals were considered as random factors. Diversity indices and both habitat availability and distance values were used as predictor variables, excluding correlating variables (Spearman’s Rho >0.7). We performed manual stepwise forward model selection using the Akaike Information Criteria (AIC) for model evaluation. Models were built using simple and squared terms, with squared terms discarded if they did not improve model prediction (Brotons et al., 2004). Following stepwise model selection, interaction terms were added based on a priori hypotheses. The best model was then used to spatially predict habitat suitability in the entire nature reserve by applying logistic transformation via logit link function to the model equation, which incorporated estimators and raster values of all final model parameters. Spatial extrapolation of probability values of habitat suitability resulted from raster calculations with the transformed model equation using the R package raster (Hijmans, 2021) and ArcGIS 10.7.1 (ESRI Inc.).

Results

Habitat selection

Forward model selection resulted in all variables entering the final model as squared terms, except for the variable availability of sand heaths and the two diversity indices food plant cover and Shannon index of plant species diversity (Table 3). Based on previous collinearity measures, number of plant species was discarded as a diversity variable in favour of the Shannon index of plant species diversity. Also, ruderal and agricultural areas were discarded because their inclusion did not improve the model. Interactions were modelled for the two dominant habitat types, sand heaths (hc) and natural grasslands (ng), each in interaction with the variables raised bogs, mires, and fens (bog), distance to dunes (dd), and food plant cover (fpc). Sand heaths were additionally interacted with shrub formations (shr) and the Shannon Index H’ (plant diversity). However, for natural grasslands these two interactions (shr and H’) were dropped in preference of a more parsimonious model due to high variances.

Table 3 Results of black grouse habitat selection calculated using generalized linear mixed models.

Diversity indices are included in (A) and excluded in (B). Squared terms are indicated as x2.

		(A) diversity indices included
AIC 8265.7
R2 0.687
Observations 13004
df 12977	(B) diversity indices excluded
AIC 8519.3
R2 0.807
Observations 16297
df 16275	
Type	Variable	Estimate	SE	p	Estimate	SE	p	
	Intercept	−5.37	0.79	<0.001	−5.53	0.49	<0.001	
Distance	dinf	13.39	0.73	<0.001	15.17	0.73	<0.001	
dinf2	−11.06	0.95	<0.001	−14.40	0.90	<0.001	
dw	8.68	0.94	<0.001	9.38	0.90	<0.001	
dw2	−9.47	1.32	<0.001	−9.50	1.25	<0.001	
dp	−2.51	0.60	<0.001	−4.33	0.55	<0.001	
dp2	5.02	0.80	<0.001	7.42	0.72	<0.001	
dd	−2.91	0.78	<0.001	−3.94	0.64	<0.001	
dd2	−1.36	0.27	<0.001	−1.18	0.25	<0.001	
Availability	hc	−3.57	0.82	<0.001	−1.29	0.41	0.001	
ng	0.28	0.84	0.735	2.21	0.53	<0.001	
ng2	−1.78	0.47	<0.001	−1.99	0.43	<0.001	
bog	−4.06	0.87	<0.001	−2.22	0.71	0.002	
bog2	3.76	0.71	<0.001	3.11	0.69	<0.001	
shr	−17.23	7.02	0.014	−16.98	6.04	0.005	
shr2	−6.23	1.03	<0.001	−4.87	1.01	<0.001	
Diversity	fpc	0.11	0.05	0.025				
H’	−0.45	0.19	0.021				
Interaction heath	hc:bog	7.19	0.86	<0.001	6.76	0.69	<0.001	
hc:shr	23.00	8.03	0.004	21.72	6.89	0.002	
hc:dd	3.76	0.95	<0.001	4.66	0.80	<0.001	
hc:fpc	−0.12	0.07	0.087				
hc:H’	1.55	0.28	<0.001				
Interaction grassland	ng:bog	−4.40	0.60	<0.001	−5.07	0.58	<0.001	
ng:dd	1.81	0.45	<0.001	1.97	0.43	<0.001	
ng:fpc	0.12	0.04	0.005				

Because inventories of plant species in the forested peripheries of the study area were often not available, diversity indices (fpc and H’) could not be calculated and were therefore excluded entirely in the final model B (Table 3B). In the other model A, diversity indices were included but data points with missing values in the surrounding forests were omitted (Table 3A).

Both models showed basically similar results for all predictor variables, with the difference that in model B all main effects and interaction effects were generally mitigated, in particular for sand heaths and natural grasslands. However, the pattern of effects was consistent between both models. The models explained increased black grouse presence in areas of high diversity of (food) plant species (fpc and H’) with a good availability of sand heaths (hc) and natural grasslands (ng) in the vicinity of raised bogs, mires and fens (bog) and with low-density shrub formations (shr) and patches of open soil/dunes (dd) (Figs. 1 and 2). As shrub formations (shr) became denser in open sand heaths, probability of black grouse presence decreased. Habitat selection by black grouse was highest at distances of 500 to 600 m from infrastructure (dinf) and dense forests (dw). It also increased significantly starting 400 m away from pastures (dp), though the vicinity to pastures still indicated marginal habitat selection. When availability of raised bogs, mires and fens was high, interaction effects showed strong positive trends in habitat suitability for sand heaths and non-dominant natural grasslands (Fig. 2).

Figure 1 Predicted probability of black grouse occurrence as a function of distance variables (left), habitat availability variables (middle) and diversity indices (right).

Distance variables are in kilometers, habitat availability is given as proportion of coverage of grouped habitats (between 0 and 1), and diversity indices are dimensionless. The main effects relate to the final model including diversity indices (Table 3A).

Figure 2 Predicted probability of black grouse occurrence as a function of interaction terms.

Availability of sand heaths hc (left column), availability of natural grasslands ng (right column). Distance variables are in kilometers, habitat availability is given as proportion of coverage of grouped habitats (between 0 and 1), and diversity indices are dimensionless. bog = raised bogs, mires, fens; shr = shrub formations; dd = distance to dunes in km; fpc = food plant cover; H’ = Shannon Index of plant diversity.

Extrapolation of habitat suitability

Spatial extrapolation of the model predictions returned several large suitable habitat patches throughout the belt of open heath and grassland (southwest to northeast), as well as in the southern peripheral heaths (Fig. 3). The westernmost patches of the reserve were predicted to be more suitable than the large eastern patch from where we gathered our black grouse movement data. Some smaller habitat patches had only low predicted suitability for black grouse (e.g., north and south of our telemetry study area). In total, only 88.3 ha or 0.37% of the entire nature reserve (235 km2) had a very high suitability (i.e., probability of black grouse presence is p > 0.75). In relation to only the open heath and grasslands, the area percentage of very high suitability was 1.9%. 285.7 ha (or 6% of open heath) had moderate to high suitability (0.5 <p < 0.75), and 512.9 ha (10.5% of open heath) had low to moderate suitability (0.25 <p < 0.5). The greatest part of the nature reserve consisted of forest (66%) which accounted for a major part of the low to unsuitable areas (226 km2; p < 0.25). However, 81.6% of the open heathland was also among these least suitable areas.

Figure 3 Spatial extrapolation of predicted habitat suitability shows the restriction of suitable black grouse habitats to extensive and heterogeneous heath areas in the Lüneburg Heath Nature Reserve.

(Left) General map of the Lüneburg Heath Nature Reserve showing the land use types within its boundaries. (Right) Predicted habitat suitability, with black lines indicating the areas within which is heath. Overview map (top) shows the location of the nature reserve in Lower Saxony (grey), Germany. Location data of the GPS-tagged grouse can be viewed in Tost et al. (2020). Map source: Tost et al. (2020).

Discussion

Habitat selection

The best habitat for black grouse was undisturbed with patches of bog, within a structurally varied natural heath or grassland not too close to dense forests. The variable distance to infrastructure (dinf) emerged as the strongest effect. This confirms previous findings on the avoidance of human disturbance (e.g., hiking trails) in the same study area as in this study (Tost et al., 2020) as well as in alpine habitats (Immitzer, Nopp-Mayr & Zohmann, 2014; Patthey et al., 2011), thus emphasizing the need for undisturbed black grouse habitats as refuge areas.

Wetland availability (bog) was particularly important for black grouse habitat selection, and the mixture of bog patches with sand heaths (hc) was a favourable habitat combination, increasing predicted probabilities of black grouse occurrence. A clear preference of black grouse for habitat compositions with moorland has been described for Scottish and northern English populations (Baines, 1994; White, Warren & Baines, 2015), where this habitat type makes up a substantial portion of areas inhabited by black grouse. In contrast, in the Lüneburg Heath Nature Reserve (dry) sand heaths and natural grasslands are the most common habitats, while there are few bogs that occupy a small area (2%). Thus, our results indicate a selective use of the few available bog patches. These findings may have important implications for conservation management not only in the nature reserve but also at the landscape scale, because most black grouse sites that went extinct in the federal state Lower Saxony until the mid-1980s were in regions with raised bogs and mires (Ludwig, Storch & Wübbenhorst, 2008), exploited for peat mining (Ludwig, Storch & Graf, 2009). Consequently, renaturation of wet habitats not only inside existing black grouse habitats but also within dispersal distance is probably one of the most important measures to strengthen and increase the black grouse metapopulation.

Black grouse chicks need abundant arthropods during their first weeks (Klaus et al., 1990; Wegge & Kastdalen, 2008). Although not investigated in this study, arthropods may be more abundant in the wetter valleys than in the higher and drier sand heaths (Baines, 1994; Patthey et al., 2011; Schweiger, Nopp-Mayr & Zohmann, 2012), which would improve offspring survival in these areas. For the mixture of bog and natural grassland (ng), the positive effect was less pronounced in our model. Here, the probability of black grouse occurrence was increased only when the availability of natural grassland was low to moderate, and the availability of bogs was high, thus supporting the assumption that wet habitats are preferred by black grouse and should be promoted by conservation management. It is possible that heather is preferred as hiding cover (Immitzer, Nopp-Mayr & Zohmann, 2014; Patthey et al., 2011; Wegge & Kastdalen, 2008) over natural grassland near bogs. Nevertheless, natural grasslands might be of importance during other phases of the black grouse life cycle, e.g., as display or nesting sites—both tagged hens nested in grassy areas.

Heterogeneity and patchiness of habitats and vegetation are key factors for black grouse habitat selection, as elaborated in several studies (Immitzer, Nopp-Mayr & Zohmann, 2014; Patthey et al., 2011; Schweiger, Nopp-Mayr & Zohmann, 2012; White, Warren & Baines, 2015). In our study area, this effect was explained in our models by both, food plant cover (fpc) and the Shannon index H’ (plant diversity) as measures of heterogeneity. Furthermore, when combining sand heaths and wet habitats in an interaction, habitat selection by black grouse was especially pronounced. The positive effect of sand heaths was strongly enforced when wet habitats became dominant. This finding should further encourage black grouse managers to create and maintain mosaics of these two important habitat types.

Dominant cover of shrub formations (shr) did not improve habitat suitability. Plots with high shrub proportions even reduced selection. However, this only applied to areas with high density of predominantly scattered shrubs. For areas with lower shrub abundance, this observation did not apply. Field observations showed that hens and cocks use solitary pines and birches as lookouts, especially during the mating season. In addition, mature trees act as an important food source during spring (Strauß et al., 2018). In fact, an adequate supply of low-density woody plants should be provided, as they promote the availability of anthills as essential chick food (Schweiger, Nopp-Mayr & Zohmann, 2012; Signorell et al., 2010; Wegge & Kastdalen, 2008). According to Wegge & Kastdalen (2008), young black grouse broods in Norwegian boreal forest preferred pine bogs with lower tree and shrub density, but higher potential predation risk over denser, bilberry-dominated forest types, which were rather used by capercaillie broods. Our study showed that dense forested areas (dw) surrounding the core habitats, but also dense woodlands in open heathland, were entirely avoided by black grouse. This likely serves the purpose of predator avoidance (Brown, Laundré & Gurung, 1999; Laundre, Hernández & Ripple, 2010) but may also be due to a ground vegetation unfavourable for mobility and feeding. In Scotland, mosaics of young forests (younger than 14 years) within moorlands are important habitats for winter and spring foraging, but also act as lek sites, breeding grounds, and shelter from predation. However, these benefits are lost in old growth forests due to the change towards a less suitable ground vegetation (White, Warren & Baines, 2015). Alpine black grouse habitats are known to span above the timberline with preference for semi-open heterogeneous patches of alpine meadows and (dwarf) shrubland, intermixed with low-density young and mature trees (Immitzer, Nopp-Mayr & Zohmann, 2014; Patthey et al., 2011; Schweiger, Nopp-Mayr & Zohmann, 2012). Given the avoidance of forest edges in the Lüneburg Heath Nature Reserve and the positive habitat suitability of forest transition zones according to previous studies, it is recommended to lighten the edge structure, away from a vertical forest edge to a gradual transition from open land to forest area, with associated understory of heather and berry bushes (Vaccinium spp. and Empetrum sp.).

Interestingly, black grouse habitat selection increased remarkably when distance to pastures (dp) was higher than 400 m. While extensive pasture farming was beneficial for black grouse in the 1950s and −60s (Ludwig, Storch & Gärtner, 2009; Ludwig, Storch & Graf, 2009), contemporary intensive pasture management with low plant diversity and high nutrient influx might be detrimental for the grouse. Despite pasture management in the nature reserve is still carried out extensively and without fertilization, these areas, as well as the surrounding heaths and natural grasslands, are affected by significant atmospheric nutrient deposition. However, pastures play only a minor role in the black grouse’s habitat mix in the Lüneburg Heath Nature Reserve. Studies in the Alps (alpine meadows) as well as in England assessed pastures rather positively (insect availability) (Baines, 1994; Patthey et al., 2011), as long as grazing was neither too intensive (Calladine, Baines & Warren, 2002) nor too light (Immitzer, Nopp-Mayr & Zohmann, 2014; Schweiger, Nopp-Mayr & Zohmann, 2012), and concealment was provided nearby (Signorell et al., 2010). Incidentally, these statements on grazing are particularly applicable to extensive sheep grazing in the open heath landscape of the nature reserve. Regarding monotonous, fenced pastures, the positive effects might only apply seasonally with corresponding vegetation height.

Our results apply primarily to spring and summer, and to a limited extent to autumn, due to the coverage period of the telemetry study. Habitat selection during autumn and winter might differ from our results. For instance, during past years, groups of black grouse have occasionally been sighted in winter on farmland in the nature reserve (extensive cultivation, e.g., buckwheat). However, farmland could not be incorporated into our model due to lack of presence data.

Extrapolation of habitat suitability

The final model accurately predicted well-suited habitats in the nature reserve’s central belt of open heath and grassland (Fig. 3), which is supported by long-term observations of grouse (Wormanns, 2021, pers. comm.). While core habitats in the western part of the nature reserve appear to be connected (relatively low fragmentation), spatial extrapolation of our models revealed a possible isolation of the core habitat in the eastern part. The vast heathland areas that lie between these two core areas are largely unsuitable habitats, due to the high density of trails, an important local road and some settlements and single buildings in this area.

Our model predicted good habitat suitability for the southernmost heathlands, but these have not been populated by black grouse for several years. We suspect that this discrepancy may relate to nearby wind turbines (Coppes et al., 2019; Coppes et al., 2020) to the east and a waste disposal facility and conventional farms to the west, adding to other disturbances within these habitats. Although several further sites of open heathland remain in the Lüneburg Heath Nature Reserve, they seem to be of poor suitability for black grouse according to model predictions. Again, this is probably due to the high density of trails (Tost et al., 2020) and the proximity to older forests. Observations during annual black grouse censuses and reports of incidental observations support this prediction. This implies that numerous considerable parts of the open heaths—mainly in the nature reserve’s periphery—are effectively inaccessible to the local population, which is instead restricted to the larger, centrally located heathland areas. These peripheral areas, however, are important for habitat connectivity with the southern adjacent military training areas and thus for dispersal between the fragmented subpopulations (Andrén, 1994; Hanski, 2008). Thinning of forest edges could increase attractiveness of such areas, provided they are undisturbed, unfragmented areas of sufficient size in the first place.

Management and research implications

There are a variety of management implications, accordingly the following actions are recommended: (1) creation of small scale habitat mosaics of heterogeneous dry heaths and natural grassland with high diversity of food plants throughout the open landscape, (2) restoration and promotion of wet habitats (mires, raised bogs) where topographically possible (White, Warren & Baines, 2015), (3) avoidance of large, monotonous heath and grassland areas, (4) reduction of regeneration stage of young, emergent trees (pioneer vegetation), but preservation of solitary trees (pines, birches, juniper) as food source and shelter (Sim et al., 2008; Patthey et al., 2011), (5) thinning of forest edges and creation of transition gradients over several hundred meters (Sim et al., 2008), (6) provision of micro patches of open soil (Klaus et al., 1990), (7) visitor guidance and enforcement of its compliance (Immitzer, Nopp-Mayr & Zohmann, 2014), promotion of environmental education with local schools (Freund et al., 2020), relocation of infrastructure to less exposed areas or landscaping to provide visual cover by vegetation, and reduction of tourism pressure if possible (Tost et al., 2020), (8) mixing pastures with habitat elements such as dunes, loose shrub formations for concealment (Signorell et al., 2010), and wet zones, (9) improvement of habitat connectivity (Andrén, 1994): consideration of fragmentation effects and thus reduction of isolation inside as well as outside the nature reserve’s borders by creation of heterogenous habitats beyond the core areas. These measures are recommended regardless of the individual interests of the various stakeholders and any resulting conflicts.

Next steps in local black grouse research should focus on microhabitats (Patthey et al., 2011) by examining vegetation species composition in detail, monitoring arthropod abundances in different habitat types (Wegge & Kastdalen, 2008), and evaluation of landscape management practices, e.g., heath mowing, sod cutting, heath burning, and sheep grazing. In addition, the overlap of black grouse habitats with those of the most common predators are currently being investigated, thus identifying conflict zones (Signorell et al., 2010). Further telemetry studies with higher sample sizes (more individuals, longer duration, shorter timing of fixes) and new study areas would be desirable. However, pragmatic reasons might complicate the realization regarding the low chances of success in catching animals in the nature reserve (only 33 individuals counted in 2021, F Stucke, 2021, pers. comm.) or on the closed military training areas. An alternative could be a scientifically supervised translocation of Swedish black grouse to the Lüneburg Heath, as it is already done in the Bavarian Rhön and in the Dutch Sallandse Heath. With no doubt, the prevailing metapopulation context is one of the central reasons that black grouse still exist in the Lüneburg Heath area. Therefore, dispersal between military training areas and the nature reserve as well as the individual roles of the five subpopulations for long-term survival of the entire metapopulation should be given special attention in future studies.

Supplemental Information

Supplemental Information 1 Classification of mapped biotope types and linkage to corresponding model variables (grouped habitat types) and Habitats Directive Codes

Click here for additional data file.

Supplemental Information 2 Code: Processing of spatial data containing rasterization of background data and point extraction for presence and background points

Code description

- Defining the extent of the study area (area of interest).

- Processing of spatial data by rasterizing biotope and vegetation mapping data, applying moving window rasterization, processing distance rasters and generating raster-stacks.

- Processing of presence data, i.e., GPS location data of 7 black grouse. Defining buffered extent for random background points and creating random points.

- Binding of presence and background data, performing point extraction on raster-stack and exporting data table (Table S1).

Click here for additional data file.

Supplemental Information 3 Code: Pre-checking for correlated variables and stepwise building of habitat suitability models

Code description

- Correlation analysis of biotope, distance and diversity variables.

- Model building and model selection. (1) Univariate GLMM, (2) Multivariate GLMM, (3) Full additive GLMM, (4) Multivariate GLMM with interactions.

- Visualization and export of final models’ effect plots.

Click here for additional data file.

Supplemental Information 4 Code: Application of the final model’s prediction to the entire area of the nature reserve Lüneburg Heath

Code description: Loading raster data (see code-file raster_processing) and the final model (Mi4c). Applying logit-transformation on model equation and performing projection/prediction with raster-stack. Calculating area of habitat suitability for 25% intervals of p-values.

Click here for additional data file.

Supplemental Information 5 Dataset for model building

Dataset containing all variables used for model building, including presence and background data, animal ID and environmental variables (availability, distance and diversity). For more information see Table S2.

Click here for additional data file.

Supplemental Information 6 Abbreviations list

Click here for additional data file.

We wish to thank the association Verein Naturschutzpark e.V. and the foundation Stiftung Naturschutzpark Lüneburger Heide for their energetic and logistic support of this project. We particularly thank M. Zimmermann, M. Sander, S. Wormanns, F. Stucke and S. Weber. Furthermore, we also thank all enthusiasts who supported and realised our fieldwork, especially J. Hindersin and A. Niebuhr. Moreover, we wish to thank K. Sandkühler from the Lower Saxony Federal Ornithological Station for contributing the numbers of annual black grouse countings. We would also like to thank O. Keuling and U. Voigt for their scientific support.

Additional Information and Declarations

Competing Interests

Author Contributions

Animal Ethics

Field Study Permissions

Data Availability

The authors declare there are no competing interests.

Daniel Tost conceived and designed the experiments, performed the experiments, analyzed the data, prepared figures and/or tables, authored or reviewed drafts of the article, and approved the final draft.

Tobias Ludwig conceived and designed the experiments, authored or reviewed drafts of the article, and approved the final draft.

Egbert Strauss conceived and designed the experiments, authored or reviewed drafts of the article, and approved the final draft.

Klaus Jung conceived and designed the experiments, authored or reviewed drafts of the article, and approved the final draft.

Ursula Siebert conceived and designed the experiments, authored or reviewed drafts of the article, and approved the final draft.

The following information was supplied relating to ethical approvals (i.e., approving body and any reference numbers):

All stages of the animal experiment were conducted under a permit from the Lower Saxony Institute for Consumer Protection and Food Safety (LAVES, Dept. 33 Animal Welfare, permit number: 33.9-42502-04-11/0364).

The following information was supplied relating to field study approvals (i.e., approving body and any reference numbers):

Field experiments were approved by the lower nature conservation authorites of the district Soltau-Fallingbostel (permit number: 09.509 N 24 - Lü 2 - 4) and district Harburg (permit number: 71 21/1.2.1-0.0 - 2011-0081 -Kr).

The following information was supplied regarding data availability:

The data and code are available in the Supplementary Files.

The environmental raw data (vegetation and biotope mapping) were provided by the Lower Saxony Water Management, Coastal Defence and Nature Conservation Agency (NLWKN) for our analyses but is not permitted to be made publicly available.

Please contact the NLWKN’s branch office (Betriebsstelle) Lüneburg (https://www.nlwkn.niedersachsen.de/startseite/wir_uber_uns/geschaftsbereiche/geschaftsbereich_4/geschaeftsbereich-iv-regionaler-naturschutz-45362.html) to apply for access to the third-party data:

-https://www.nlwkn.niedersachsen.de/live/contact_form.php?a=46111&c=1999

-https://www.nlwkn.niedersachsen.de/live/contact_form.php?a=46111&c=2074

- +49 (0)4131 / 2209-202

- +49 (0)4131 / 2209-219

The third-party data is called “digital Data (Text-E-, Datenbank und shape-Entwurf zur Basiserfassung Lüneburger Heide/FFH Nr.070)”.

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
