# Peer review of "Habitat selection of black grouse in an isolated population in northern Germany—the importance of mixing dry and wet habitats"

_PeerJ, doi:10.7717/peerj.14161_

## Round 0.1 · original submission · Minor Revisions

The two reviewers concurred that revisions are necessary, and I agree. Please read their comments carefully and thoroughly. I have also added an annotated copy of your manuscript. I attempted to correct some of the writing, as well as make several suggestions for content, figures and tables.

I am particularly concerned about the way you worded your "hypotheses" that seem to me to simply say you are "predicting" what is already known about Black Grouse habitat selection. You need to be more precise in your wording so that your null, or alternative, hypothesis is possible, rather than false by definition. Then, in the text where you describe your results, you should place that clearly in the context of the new hypotheses.

I hope you find all of these suggestions to be helpful in your revisions.

Reviewer 1 ·

Basic reporting

I suggest including a few references regarding the biology of the species (habitat preferences) in order to better link the ideas on the introduction to the objectives of the study.

Experimental design

No comments.

Validity of the findings

The results area very meaningful. However I do not believe the conclusions are according to their findings. I suggest considering concluding about:
1. are the occurrence probability values high or low for the black grouse compared to other populations or other species?
2. the importance of the reserve for the black grouse within other areas which actually show higher habitat suitability values.

These areas should be better described, for there are no information regarding them. Are they part of the important military areas? Are they within private lands? Do they have historical or current records of the species etc.

Additional comments

No comments.

Annotated reviews are not available for download in order to protect the identity of reviewers who chose to remain anonymous.

·

Basic reporting

Most parts of the text are clear and well-written. Some points of confusion may be easily clarified, and I include specific comments (additional comments) that may help improve the text.

The literature cited is focused on the study species. The text may benefit from the citations of more broad studies or information from other species that may be relevant to the topics in the introduction. Some points in the discussion also need citations.

The understanding of the raw data included as a supplemental file may be improved with a metadata file that explains the variables and their names in the data table.

The article is written in a very objective way and the results reflect most of what is described in the methods and hypothesis. The introduction, methods, and some parts of the discussion may be improved though.

Experimental design

The methods used to sample and analyze the data seem adequate, although other approaches could be better suitable for increasing the prediction power of models (such as machine learning algorithms with training and testing datasets, or null models testing). The random sampling of background points may have worked fine with logistic mixed models due to the fact that the species does not prefer the most common type of habitat in the landscape or/and that the different types of biotopes were present in the background sampling areas for most individuals.

The description of methods can be improved, including brief descriptions of the variables used for biotope classification and grouping. Also, a simple description of how the models were used to predict habitat suitability is necessary.

Validity of the findings

Some parts of the discussion and conclusions need to be better supported by including citations (see additional comments). The article is overall well structured but would benefit from a better description and contextualization of the importance of predictions of habitat suitability.

Additional comments

Introduction

lines 39-79: The introduction could benefit from a more broad explanation of aspects of habitat selection and habitat use and how anthropogenic changes can affect wildlife (e.g., information from studies developed with other species and more general information about the topics). This is just a suggestion though, as the authors can opt to keep it as it is: short and focused on the study species only.

lines 78-79: Predicting suitable areas is extremely important in the context of the study. The objective of predicting habitat suitability could be better explored in the introduction, as well as better described in the methods (also, see more comments in the results).


Methods

line 116: I believe you sampled locations every three hours during the entire sampling period, with the first at 01:00 and the last at 23:00 daily, is that correct? Please rewrite for clarification.

lines 125-139: What are the main variables used for biotope mapping? A short description of the environmental variables used for biotope mapping will increase clarity.

lines 144-147: The grouping reduced the number of biotopes from 166 to around 50? Please clarify the number of resulting groups in the text. Also, the text will be much improved if a short description of the attributes used to group the biotopes is included in this part of the text.

lines 150-151: Can a cell contain more than one biotope? Please clarify.


Results

line 180: The terms 'resource selection' and 'habitat selection' may be interchangeable in several occasions. Is there any particular reason to choose one over the other here or in other parts of the text? If not, I suggest you opt for one of them and use it throughout, or at least in the sections and subsections titles.

lines 210-223: This may be one of the main results of your study, as it can be used to build conservation strategies for the species in the study region. Although predicting occurrence may be a consequence of the fitted models, you must describe the steps taken to predict habitat suitability in the methods and also explore this objective better in the introduction.

Discussion

lines 344-359: Although this part of the discussion reflects specific suggestions for the populations in the study region, I believe it would benefit from examples from other cases, even with other species or more general studies that show that the recommendations are, in fact, efficient for conservation. Is it possible to include some citations to refer to other studies that may support your claims that these recommendations are in fact efficient?

lines 362-375: Many sentences in this paragraph need citations, including those that refer to results or data from other studies.

---

## Round 0.2 · Minor Revisions

While we appreciate your efforts in improving the text, I found some additional details that I think you might clarify. Rereading some of the reviewer's comments, I see that both of them asked for clarification on some of these same issues. I would appreciate it if you would re-read their reviews along with my comments in the annotated manuscript. This is to ensure that you clarify any confusing points, and improve the English a bit more.

Please note in my annotated manuscript that all marks of the text have comments associated with them. These are all suggestions that should help improve the manuscript and clarify any outstanding issues. My suggestions are not absolute, of course, and if you find a better way to clarify where I provide suggestions, feel free to use your better judgment.

---

## Round 0.3 · accepted · Accept

Thank you for taking the time to revise your manuscript - the improvements are apparent. Because the original reviewers suggested minor revisions, this time I did not send it back to them, because I have also read and reviewed the manuscript and I think you satisfied their comments as well.

The manuscript is essentially ready for publication, but I would like to call your attention to your tables. Please look at some examples of tables and try to make them more readable. Here are some suggestions.

Table 1. Center text in cells instead of left justified. Use "-" instead of "." to separate days-months-years. Remove the column "age" and in the legend just indicate with an asterisk (or your choice) that all are adults except 1207 that was caught as a yearling. Get rid of lines, except at the bottom edge of table and under the column headings.

Table 2. In the first column, Type, and first item (Distance), write (km) there, instead of repeating it in the third column. In the third column, now the word "Distance" is superfluous because it's in the first column. So, you can just write "To infrasctructure" or "To forest." And, in "availability" you did not include "units" as you did in distance. Perhaps you should remind the reader of how "availablity" is measured. Get rid of lines as in table 1.

Table 3. You have "< 0.0001" so many times as to be essentially unnecessary. You can eliminate those to P columns and simply write in the legend that "unless otherwise stated in parentheses after the variable, all P < 0.001." Then, for example, in the availability variables, you could write ng (0.735), and so on. This would free up two columns and make the table more readable. And, lines as in tables 1 and 2.

Of course, if the instructions to the authors require the lines in the tables, then you can ignore that part of my comments. Please double-check all formating requirements.